# The Challenges of Gastric Cancer Surgery during the COVID-19 Pandemic

**DOI:** 10.3390/healthcare11131903

**Published:** 2023-06-30

**Authors:** Catalin Vladut Ionut Feier, Alaviana Monique Faur, Calin Muntean, Andiana Blidari, Oana Elena Contes, Diana Raluca Streinu, Sorin Olariu

**Affiliations:** 1First Discipline of Surgery, Department X-Surgery, “Victor Babes” University of Medicine and Pharmacy, 2 E. Murgu Sq., 300041 Timisoara, Romania; catalin.feier@umft.ro (C.V.I.F.); olariu.sorin@umft.ro (S.O.); 2First Surgery Clinic, “Pius Brinzeu” Clinical Emergency Hospital, 300723 Timisoara, Romania; diana.streinu@umft.ro; 3Faculty of Medicine, “Victor Babes” University of Medicine and Pharmacy, 300041 Timisoara, Romania; 4Medical Informatics and Biostatistics, Department III-Functional Sciences, “Victor Babes” University of Medicine and Pharmacy, 2 E. Murgu Sq., 300041 Timisoara, Romania; cmuntean@umft.ro; 5Oncology, Department IX-Surgery, “Victor Babes” University of Medicine and Pharmacy, 2 E. Murgu Sq., 300041 Timisoara, Romania; andiana.blidari@umft.ro (A.B.); oana.conte@umft.ro (O.E.C.)

**Keywords:** COVID-19 pandemic, gastric cancer surgery, length of hospitalization, surgical postponement

## Abstract

The aim of this study was to quantify the impact of the COVID-19 pandemic on the surgical treatment of patients with gastric cancer. Data from patients undergoing surgery for gastric cancer during the pandemic were analyzed and the results obtained were compared with the corresponding periods of 2016–2017 and 2018–2019. Various parameters were taken into consideration and their dynamics highlight significant changes in the pandemic year compared with the two pre-pandemic periods. Statistical analysis revealed a marked decrease in the number of surgeries performed during the pandemic (*p* < 0.001). Severe prognostic factors for gastric cancer, including weight loss and upper gastrointestinal hemorrhage, were associated with an increased number of postoperative fistulas, while emesis was statistically correlated with a more advanced cancer stage (*p* < 0.011). There was also a reduction in the total duration of hospitalization (*p* = 0.044) and postoperative hospitalization (*p* = 0.047); moreover, the mean duration of surgical intervention was higher during the pandemic (*p* = 0.044). These findings provide evidence for the significant changes in clinical and therapeutic strategies applied to patients undergoing surgery for gastric cancer during the study period. The ongoing pandemic has exerted a substantial and complex impact, the full extent of which remains yet to be fully comprehended.

## 1. Introduction

The COVID-19 pandemic has presented significant challenges to healthcare systems on a global scale [1]. Originating in Wuhan, China, towards the end of 2019, the outbreak of the SARS-CoV-2 virus rapidly disseminated, necessitating crucial adaptations in treatment protocols, particularly for patients with oncological conditions who required surgical interventions [2,3]. The COVID-19 pandemic has had a significant impact on admission patterns for patients treated for gastric cancer. Firstly, disruptions in healthcare systems resulted in reduced access to screening programs and diagnostic procedures, leading to delayed diagnosis and patients being admitted at later stages of the disease. This delay in diagnosis can adversely affect treatment outcomes, increase the complexity of surgical procedures, and necessitate more complex treatments. Secondly, modifications in healthcare delivery, such as prioritizing urgent cases and postponing elective surgeries, have affected admission patterns for gastric cancer patients. This has led to rescheduled surgeries and altered treatment plans to minimize the risk of COVID-19 transmission. Additionally, fear of contracting the virus and overwhelming healthcare systems has led to reduced healthcare-seeking behavior, decreasing the number of admissions for gastric cancer treatment. Finally, variations in healthcare capacity due to the strain caused by the pandemic, including limited hospital resources, have impacted admission patterns, with hospitals dealing with COVID-19 cases facing limitations in admitting and treating non-emergency cases [4,5].

Gastric cancer ranks fifth in terms of global cancer incidence and represents a significant cause of mortality. According to GLOBOCAN 2020 estimates, 1.1 million new cases of gastric cancer were diagnosed, with the disease ranking fourth in terms of mortality [6]. The treatment of this pathology is complex, and the consequences generated by the pandemic in terms of the management of these patients are not yet fully known.

This study endeavors to address the prevailing knowledge gap by conducting a comprehensive analysis and interpretation of the clinical and pathological aspects pertaining to patients undergoing gastric surgery during the unprecedented COVID-19 pandemic. With a focus on the unique timeframe, this investigation aims to meticulously monitor and evaluate the dynamic changes observed in various parameters, in comparison to previous periods. By examining patients receiving surgical treatment for gastric cancer at the First Clinic of Surgery at “Pius Brinzeu” Emergency County Clinical University Hospital in Timisoara, Romania, this study strives to elucidate the profound impact of the pandemic on these patients.

## 2. Materials and Methods

This study examines a cohort of patients who underwent surgical intervention for the treatment of gastric cancer at the First General Surgery Clinic of Timisoara County Emergency Clinic Hospital, Pius Brinzeu, which is one of the hospital’s three surgery clinics. The data from 137 patients who underwent surgery between 26 February 2020 and 1 October 2021 were considered and analyzed. The results were then compared with the corresponding periods of 2018–2019 and 2016–2017. It is worth noting that 26 February 2020 marks the first confirmed case of COVID-19 in Romania.

To conduct this study, several inclusion criteria were established. These criteria encompassed patients who received surgical intervention for gastric cancer treatment within the specified timeframe and obtained a malignant result from the morphopathological examination of the resected specimen. Additionally, during the pandemic, supplementary inclusion criteria were introduced to ensure a comprehensive evaluation. The study only included patients who were not actively infected with COVID-19 at the time of seeking medical care or during their hospitalization, and who had no prior history of SARS-CoV-2 virus infection. Only patients who tested negative for SARS-CoV-2 in the RT-PCR test conducted upon hospitalization were included.

Furthermore, during the pandemic, patients displaying characteristic symptoms of COVID-19 within seven days prior to their hospital admission were excluded from the study. Ethical approval for data collection was obtained from the Hospital Commission (No. 384/08.03.2023).

Multiple parameters were investigated, including sex, age, place of residence, presence of upper gastrointestinal hemorrhage, emesis, more than 10% decline in body weight, and whether the patient underwent emergency or elective surgery. The Charlson index was utilized to evaluate patient comorbidities. The degree of anemia was analyzed based on the hemoglobin level (grams per deciliter-Hb/dl) as well as the patients’ need for postoperative erythrocyte concentrate transfusions. Surgical interventions were categorized into three major groups: total gastrectomy, subtotal gastrectomy, and other types of interventions, including both palliative surgeries (e.g., jejunostomy, gastrostomy) and extensive resections involving other organs. The study also examined and analyzed the recurrence rate.

The duration of surgical intervention, the intention of curative/palliative radicality, and the occurrence of postoperative complications were also analyzed. Variations in both clinical and pathological T staging (tumor invasion), the number of lymph nodes with metastases (N), and the presence or absence of metastases (M) were analyzed. Regarding the duration of patients’ hospital stays, this study analyzed three distinct periods: the length of preoperative hospitalization, the length of postoperative hospitalization, and the total length of hospitalization. Lastly, the discharge status of patients throughout the three periods (deceased/cured) was analyzed.

For statistical analysis and interpretation of the obtained results, IBM SPSS Statistics software for Windows (IBM, Armonk, NY, USA) was utilized. The analysis included calculating descriptive statistics, such as measures of central tendency and dispersion, for numerical variables. Categorical variables were examined by constructing frequency tables and percentages. To highlight statistical differences between study variables, the following tests were employed: the ANOVA test for continuous variables, and the chi-square test for differences in proportions between variables. The Pearson correlation coefficient was used to identify correlations among the study variables. Cox regression analysis was conducted to evaluate the influence of specific variables on the risk of postoperative mortality while adjusting for confounding factors. The statistical significance level was set at a *p*-value threshold of less than 0.05.

## 3. Results

To conduct this study, data from patients who underwent surgical intervention for gastric cancer treatment between 26 February 2020 and 1 October 2021 were analyzed, and the results obtained were compared with the same period in 2016–2017 and 2018–2019.

### 3.1. Patient Demographics and Clinical Manifestations

The data of 137 patients who met the inclusion and exclusion criteria of the study were considered. During the pandemic period, 28 surgical interventions (20.4%) were performed, while in the period of 2018–2019, there were 50 surgeries (36.5%), and in the initial period of the study, 59 surgical interventions were conducted (43.1%). The statistical analysis and application of the chi-square test to detect differences in proportions among the three periods resulted in a *p*-value of <0.001.

The age, sex, and environment of origin are presented in Table 1.

Analyzing the initial period of the study individually with the pandemic period, a significant decrease in the proportion of patients from rural areas is reported, from 43.1% to 21.4%. This decrease was statistically significant, as indicated by the chi-square test with a *p*-value of 0.05.

The rate of patients requiring emergency surgical intervention for gastric cancer treatment increased from 42.4% in the period of 2016–2017 and 52% in the second period to 60.7% during the pandemic. After statistical analysis and the application of the chi-square test, a *p*-value of 0.255 was obtained, indicating no significant difference among the three study periods.

The clinical manifestations of patients throughout the 3 periods are presented in Table 2.

Analyzing the pandemic period and the first period of the study, the proportion of patients with emesis upon admission is 28.6% compared to 8.5% in the 2016–2017 period. Statistical analysis and the application of the chi-square test yielded a *p*-value of 0.034, indicating significant differences between the two periods.

Taking into consideration the entire cohort, statistically significant associations were observed between the presence of body weight decline and the occurrence of postoperative complications such as fistula (*p* = 0.02), T stage (*p* = 0.02), and postoperative hospitalization duration (*p* = 0.023). The average duration of hospitalization for patients with body weight decline was 20.76 days compared to 16.82 days for others. Statistical analysis revealed an association between these two variables, generating a *p*-value of 0.042.

Regarding upper gastrointestinal hemorrhage, significant associations were found between its occurrence and the presence of postoperative complications such as fistula (*p* = 0.011).

The presence of emesis was significantly associated with T stage (*p* = 0.005), N stage (*p* = 0.012), M stage (*p* = 0.006), and cancer stage (*p* = 0.011).

The mean value of the Charlson Comorbidity Index ranged from 4.74 ± 2.04 (2016–2017) and 5.44 ± 2.49 (2018–2019) to 6.0 ± 2.88 (during the pandemic period). Thus, statistical analysis yielded a *p*-value of 0.063, confirming no significant differences between the three periods. Nevertheless, upon analyzing the first period of the study compared with the pandemic, a significant increase in the average value of the index was observed (*p* = 0.024).

### 3.2. Key Factors and Outcomes

The degree of anemia, type of surgical intervention performed, postoperative complications (fistula), transfusion, and recurrence were examined and are presented in Table 3.

A significant increase in recurrence was observed in the pandemic period of the study compared to the 2018–2019 period (*p* = 0.039). When comparing the periods 2016–2017 and the pandemic period, the differences were not statistically significant (*p* = 0.44).

The statistical analysis did not show any significant differences in the proportion of patients requiring postoperative transfusions across the three periods. Similarly, when comparing the 2018–2019 period with the pandemic period, there was no statistically significant difference observed (*p* = 0.391). However, it is worth highlighting that a noteworthy *p*-value of 0.038 was obtained when comparing the first period with the 2020–2021 period, indicating significant differences between these two specific periods regarding this aspect.

Furthermore, an association was observed between the duration of the surgical intervention and the need for postoperative transfusions (*p* = 0.029). Patients undergoing longer surgical procedures (M = 206.43 min) had a higher likelihood of requiring transfusions compared to those who did not require transfusions (M = 175.88 min).

Palliative surgical interventions were applied in 18 cases (31.6%) during the initial study period, in 19 cases (38%) in the period 2018–2019, and in 10 cases (35.7%) during the pandemic period. Applying the chi-square test to examine the differences in the proportion of patients undergoing such interventions across the three periods resulted in a *p*-value of 0.780.

The TNM variation and the cancer stage throughout the three periods are presented in Table 4.

In the pandemic period, six patients (21.4%) died postoperatively. In the first two periods, nine patients died (representing 15.3% and 18%, respectively), but the differences between the three periods were not statistically significant. The statistical analysis and the application of the chi-square test yielded a *p*-value of 0.847.

### 3.3. Periods of Hospitalization and Duration of Surgery

Table 5 presents the variations in the duration of surgery, preoperative and postoperative hospitalization lengths, as well as the total length of hospitalization.

While there were no statistically significant differences observed in hospitalization durations among the three periods, several noteworthy findings should be mentioned. The statistical analysis revealed a significant decrease in both the average total hospitalization duration (*p* = 0.044) and the average postoperative hospitalization duration (*p* = 0.047) during the pandemic compared to the 2016–2017 period.

Across the three periods, a significant positive correlation was found between the duration of the intervention and the total duration of hospitalization (r = 0.210, *p* = 0.016), as well as the duration of postoperative hospitalization (r = 0.238, *p* = 0.006).

### 3.4. Risk Factor Analysis

The risk of postoperative mortality in relation to the duration of postoperative hospitalization was examined using a Cox regression model taking into consideration all patients. The resulting model is statistically significant (*p* = 0.001), indicating that the set of investigated variables has predictive power in assessing the risk of death at discharge. The variables identified with significant predictive power are the Charlson index (*p* = 0.032) and the type of surgical intervention (*p* = 0.012).

The hazard ratio (HR) for the Charlson index is 1.368, with a confidence interval (CI) ranging from 1.027 to 1.822. For the type of surgical intervention, the HR is 0.285, with a CI of (0.107, 0.756). These findings suggest that a one-point increase in the Charlson index increases the risk of death by 1.368, and total gastrectomy carries the highest risk of postoperative death.

## 4. Discussion

This study aimed to evaluate the impact of the COVID-19 pandemic on the treatment and management of patients with gastric cancer. It is well known that oncology patients have been neglected during the pandemic, with elective oncological surgeries being postponed in the early stages. Due to the limited information about the novel coronavirus, its high transmissibility rate, and its aggressive nature, especially in patients with comorbidities, healthcare systems worldwide have redirected a significant portion of their material and human resources toward managing these patients, while other pathologies have been neglected [4,7]. The restrictions imposed by authorities, along with their recommendations to visit healthcare facilities only in case of emergencies, as well as the psychosocial factors affecting patients (fear of contracting the novel coronavirus), have led to a delay in their presentation and, consequently, the initiation of treatment [8,9]. It should be noted that patients with gastric neoplasms are particularly susceptible to and have a higher risk of developing COVID-19, mainly due to their compromised immune status [10].

The findings revealed a significant and statistically validated decrease (*p* < 0.001) in the number of surgical procedures performed during the pandemic. Specifically, there was a remarkable 52.54% reduction compared with the initial study period and a significant 44% decrease compared with the 2018–2019 period. These findings align with global reports of substantial declines in surgical interventions, ranging from 17% to 63% in hospitals across India, and a notable 50% decrease in a Tokyo hospital. Similarly, Italy also reported a significant 30% reduction [4,11,12,13,14,15,16].

This situation has led to delays in diagnosis and initiation of treatment, significantly impacting the prognosis of patients. One contributing factor to this issue has been a notable decrease in gastric cancer screening during the pandemic. Italy reported a reduction of 68%, while Hong Kong experienced a decrease of 37%; notably, the Japanese Society of Gastroenterological Endoscopy strongly recommended the postponement or cancellation of endoscopic procedures during the initial phase of the pandemic, while other studies reported declines up to 80% [7,17,18,19,20,21,22]. These findings highlight the far-reaching consequences of reduced screening efforts, further emphasizing the urgent need to address the impact of the pandemic on timely cancer detection and intervention.

During the pandemic, patients from rural areas presented at hospitals for surgical interventions in significantly lower proportions (*p* = 0.05). A considerable decrease of nearly 50% compared to the initial period and 37% compared to the 2018–2019 period was observed. Probably, patients from rural areas typically exhibit more hesitancy and a preference to postpone hospital visits. Furthermore, their lack of information and reluctance to visit healthcare facilities have led to delayed hospital presentations. These findings underscore the need for targeted educational initiatives and improved communication to address the unique challenges faced by rural populations, ensuring timely access to necessary healthcare services and interventions [23,24].

Alongside the significant decrease in the number of surgical interventions performed, there was a decline in the number of elective surgical procedures for this condition. During the pandemic, the proportion of patients undergoing emergency surgery increased to over 60% compared to 42.4% in the 2016–2017 period and 52% in the 2018–2019 period. This trend can be ultimately justified by a higher proportion of patients presenting with upper gastrointestinal hemorrhage 28.6% (pandemic period) compared to 18% (2018–2019) even though, in the first period of the study, the percentage (27.1%) was similar to that of the pandemic period. In addition to these, a significantly increased proportion (*p* = 0.032) of patients experienced emesis. Furthermore, this study reveals a statistically significant association between the presence of emesis and unfavorable prognostic factors such as T stage (*p* = 0.005), N stage (*p* = 0.012), M stage (*p* = 0.006), and cancer stage (*p* = 0.011).

Furthermore, it was observed that the presence of significant weight loss, along with upper gastrointestinal hemorrhage, was statistically significantly associated with the occurrence of anastomotic fistula (*p* = 0.02, *p* = 0.011) as a postoperative complication. Gastric cancer is a chronic debilitating disease associated with malnutrition, compromised immunity, anemia, and other comorbidities [25]. Additionally, studies have shown that patients presented at a significantly higher proportion with severe symptoms during the pandemic (*p* < 0.001) [4,13,14,26]. In this study, no statistically significant differences were found between the three periods regarding upper gastrointestinal hemorrhage, decline in body weight, and anemia.

Last but not least, patients undergoing surgery during the pandemic exhibited a statistically significant higher mean Charlson index (*p* = 0.024) compared to 2016–2017, with an average of 6.0 ± 2.88 compared to 4.74 ± 2.04. Although there was no statistically significant difference between the pandemic period and the 2018–2019 period when comparing the mean Charlson index, it is worth noting that an increase from 5.44 ± 2.49 to 6.0 ± 2.88 was observed in the most recent period. 

In addition to this, a statistically significantly higher proportion of patients experiencing recurrence was observed (*p* < 0.05) between the 2018–2019 and the pandemic period, with an increase from 4% to 17.9%. This result should be interpreted with caution, taking into consideration the sample size of the patient cohort and the fact that when comparing all three periods, the result does not generate statistically significant differences between them.

Regarding unfavorable prognostic factors such as the TNM stage during the pandemic, all patients had a T3 or T4 stage, with the proportion of patients in stage III or IV being nearly 90%. In previous periods, this percentage was 68% (2018–2019) and 66.7% (2016–2017). Considering these aspects, the increase in average surgical intervention duration during the pandemic is justified. Due to the more advanced T stage and overall cancer stage, surgeons had to perform more complex and extensive surgeries, resulting in longer surgical procedures. This increase in surgical duration has also been reported in the specialized literature, in a study conducted in Turkey where the mean duration increased from 169 min to 247 min (*p* < 0.001) [12,27].

The literature indicates a decrease in the proportion of patients in stages I or II and an increase in those in stages III or IV [16]. This trend also applies to tumor invasion, where an increase in the proportion of patients with T3 or T4 has been observed [4,28]. Considering that the differences are not statistically significant, these increases can be justified by the delayed surgical treatment due to imposed restrictions and patients’ fear of visiting hospitals during this period.

One of the goals of surgeons was to minimize the length of time patients spent in the hospital. Therefore, this study demonstrates a significant decrease between the 2016–2017 period and the pandemic in terms of total hospitalization duration (*p* = 0.044), postoperative hospital stay (*p* = 0.047), and a 16% decrease in preoperative hospitalization duration. Although there are no significant differences between the 2018–2019 and the pandemic period, it is worth mentioning the shortening of the mean duration of these periods during the pandemic. In addition, the lack of significant changes in preoperative hospitalization duration during the pandemic can be attributed to the 24-h isolation period for patients until the results of the RT-PCR test for SARS-CoV-2 detection were obtained, following the epidemiological norms implemented by the clinic where the study was conducted. Surgeons preferred to shorten the hospital stay to minimize the risk of patients contracting the novel coronavirus. Therefore, patients were discharged once they had resumed intestinal transit, did not exhibit complications, and were hemodynamically stable.

The existing literature presents contradictory findings in this regard. Some institutions have opted to isolate patients for a period of 3 to 5 days prior to surgery during the pandemic, aiming to minimize the risk of COVID-19 infection or a false negative result on RT-PCR testing. Regarding the duration of postoperative hospitalization, it averaged around 10–15 days during this period, with the goal of minimizing contact with the hospital environment and reducing the risk of infection as much as possible [9,27,28].

After conducting an in-depth statistical analysis, significant correlations were found between the duration of surgical intervention and postoperative hospitalization duration (r = 0.238, *p* = 0.006), as well as between the latter and the total duration of hospitalization (r = 0.210, *p* = 0.016). These correlations can be attributed to the more advanced stages of invasion and stage. Patients who underwent prolonged surgeries and required extended postoperative monitoring experienced slower recovery and needed additional supervision.

A comprehensive analysis was conducted to assess the risk factor associated with postoperative mortality in the entire study cohort, utilizing Cox regression. The findings demonstrated that total gastrectomy, in conjunction with the Charlson Comorbidity Index, was significantly associated with an increased rate of postoperative mortality. Notably, the mean Charlson index exhibited a significant rise during the pandemic period (in particular compared to the 2016–2017 period), suggesting a heightened risk of mortality. This observation is further substantiated by the elevated postoperative mortality rate of 21.4% during the pandemic, compared to 15.3% and 18% in the preceding periods. Additionally, it is noteworthy that patients during the pandemic solely presented with stage T3 and T4 tumors, accompanied by a considerably higher incidence of emesis, which has been consistently linked to an unfavorable prognosis. Consequently, these factors collectively contributed to an upsurge in the mortality rate during this challenging period.

Regarding patient prognosis, studies in the literature indicate a slight increase in postoperative mortality rate, although not statistically significant [29]. The delay of surgical treatment by 2, 4, 6, or 8 weeks does not impact patient prognosis, particularly in stages I or II [9,13,16,28,30].

### Study Limitations

This study has several limitations that should be considered. Firstly, it was conducted in a single clinic of a University Tertiary hospital in Romania, and therefore the size of the patient cohort should be taken into consideration, which limits the generalizability of the findings to a national level. Secondly, only patients without prior or concurrent COVID-19 infection were included in the study, focusing solely on the surgical management of gastric cancer patients and not the impact of SARS-CoV-2 infection on these individuals.

Another limitation of the study is the potential for selection bias due to previous SARS-CoV-2 infection and multiple waves of infection. This could lead to a disproportionate representation of lower-income individuals unable to work remotely, affecting generalizability and representativeness. However, despite these limitations, the study sheds light on the consequences of this challenging situation, which aligns with the existing literature in the field. Further in-depth research and comprehensive studies are necessary to accurately determine the influence of this period on the prognosis of these patients.

## 5. Conclusions

The surgical landscape has been significantly impacted by the COVID-19 pandemic. This study highlights the dramatic changes that have occurred in the surgical treatment and management of patients with gastric cancer. It is well known that surgery remains a crucial tool in the treatment of gastric cancer. However, during this pandemic, a significant decrease in the number of surgical interventions performed and an increased duration of such procedures have been observed. These aspects can be attributed to factors including reduced screening, patient anxiety, and government-imposed restrictions.

However, it is noteworthy to acknowledge that the results of this research conducted in a single clinic may not fully capture the overall impact of the pandemic on gastric cancer patients. It is plausible that a larger number of undiagnosed patients existed during this period, which could potentially result in increased mortality rates in the future. Therefore, it is of utmost importance to exercise careful management of these patients in the upcoming period, ensuring prompt surgical treatment for those who have postponed their interventions or recently been referred to surgical services.

## Figures and Tables

**Table 1 healthcare-11-01903-t001:** Age, sex, and environment of origin of the patients.

Variables	2016–2017	2018–2019	2020–2021	*p*
Sex				0.941
Men	44 (74.6%)	36 (72%)	21 (75%)	
Women	15 (25.4%)	14 (28%)	7 (25%)	
Age (years, M ± SD)	66.86 ± 11.14	62.68 ± 9.98	64.82 ± 11.49	0.135
Environment				0.139
Urban	33 (56.9%)	33 (66%)	22 (78.6%)	
Rural	25 (43.1%)	17 (34%)	6 (21.4%)	

**Table 2 healthcare-11-01903-t002:** Clinical manifestations of patients.

Variables	2016–2017	2018–2019	2020–2021	*p*
More than 10% decline in body weight	25 (42.4%)	11 (22%)	9 (32.1%)	0.078
Upper gastrointestinal hemorrhage	16 (27.1%)	9 (18%)	8 (28.6%)	0.445
Emesis	5 (8.5%)	12 (24%)	8 (28.6%)	0.032

**Table 3 healthcare-11-01903-t003:** Degree of anemia, type of surgical intervention, postoperative complications (fistula) rate, transfusions, and recurrence.

Variables	2016–2017	2018–2019	2020–2021	*p*
Anemia				0.259
Normal (Hb: >11 g/dL)	16 (29.6%)	18 (36.7%)	7 (25%)	
Grade I (Hb: 9.5–10.9 g/dL)	16 (29.6%)	9 (18.4%)	3 (10.7%)	
Grade II (Hb: 8–9.4 g/dL)	11 (20.4%)	12 (24.5%)	12 (42.9%)	
Grade III (Hb: 6.5–7.9 g/dL)	10 (18.5%)	9 (18.4%)	4 (14.3%)	
Grade IV (Hb: <6.5 g/dL)	1 (1.9%)	1 (2.0%)	2 (7.1%)	
Type of surgery				0.135
Total gastrectomy	12 (20.3%)	8 (16%)	4 (14.3%)	
Subtotal gastrectomy	34 (57.6%)	30 (60%)	15 (53.6%)	
Other interventions	13 (22.1%)	12 (24%)	9 (32.1%)	
Postoperative complication (fistula)	7 (11.9%)	9 (18%)	4 (14.3%)	0.092
Postoperative transfusion	26 (44.1%)	29 (58%)	19 (67.9%)	0.089
Recurrent disease				0.139
Yes	7 (11.9%)	2 (4%)	5 (17.9%)	
No	52 (88.1%)	48 (96%)	23 (82.1%)	

**Table 4 healthcare-11-01903-t004:** TNM and stage of cancer during the 3 periods.

Variables	2016–2017	2018–2019	2020–2021	*p*
T1	1 (1.9%)	1 (2%)	0 (0%)	0.479
T2	7 (13.0%)	4 (8%)	0 (0%)	
T3	14 (25.9%)	13 (26%)	6 (21.4%)	
T4	32 (59.3%)	32 (64%)	22 (78.6%)	
N0	9 (16.7%)	5 (10%)	2 (7.1%)	0.330
N1	12 (22.2%)	10 (20%)	6 (21.4%)	
N2	11 (20.4%)	4 (8%)	4 (14.3%)	
N3	22 (40.7%)	31 (62%)	16 (57.1%)	
M0	40 (74.1%)	32 (64%)	17 (60.7%)	0.381
M1	14 (25.9%)	18 (36%)	11 (39.3%)	
Stage				0.258
I	3 (5.2%)	1 (2%)	0 (0%)	
II	16 (28.1%)	15 (30%)	3 (10.7%)	
III	22 (38.6%)	16 (32%)	14 (50.0%)	
IV	16 (28.1%)	18 (36%)	11 (39.3%)	

**Table 5 healthcare-11-01903-t005:** Variations in surgical duration and hospitalization lengths.

Variables	2016–2017	2018–2019	2020–2021	*p*
Duration of surgery (min., M ± SD)	165.61 ± 72.66	196.55 ± 76.0	240.64 ± 81.44	<0.001
Preoperative hospitalization (days, M ± SD)	4.30 ± 2.74	4.06 ± 2.35	3.61 ± 1.97	0.477
Postoperative hospitalization (days, M ± SD)	16.20 ± 11.44	13.0 ± 7.94	11.75 ± 8.39	0.083
Total hospitalization (days, M ± SD)	20.31 ± 12.90	17.06 ± 8.49	15.36 ± 8.21	0.089

## Data Availability

The datasets used and/or analyzed during the current study are available from the corresponding author upon reasonable request.

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
