# Peer review of "The Challenges of Gastric Cancer Surgery during the COVID-19 Pandemic"

_healthcare, 2023, doi:10.3390/healthcare11131903_

Round 1

Reviewer 1 Report

This manuscript aims to evaluated the impact of Covid-19 pandemic on the surgical treatment of patients with gastric cancer (GC), by comparing this period with the corresponding periods of 2016–2017 and 2018–2019. There is an awareness among the entire medical community about the changes in the treatment of cancer patients during the Covid-19 pandemic, however it is urgent to quantify this impact more accurately. This work is an important step in that evaluation and highlights several aspects that have undergone changes and had an impact on the surgical treatment of GC patients. Works like this are of the utmost importance, so that in future crisis situations, more effective measures can be taken to prevent the treatment of cancer patients from being affected.

This is a well-structured article, regarding a trending topic, and with mostly recent reference citations.  There is a clear methodology and results are interesting. The fact of having to periods of comparison prior to the COVID-19 pandemic is a point of strength in this work, allowing more solid conclusions. However, sometimes authors don’t mention the results for all periods (please see below the specific comments), which would be important in order to have more accurate results and conclusions. Regarding the results topic, the tables properly show the data and are easy to interpret. Regarding discussion and conclusion, the authors rise good points of discussion and reinforce their conclusion most of the time with relevant Literature references. However, authors should be more careful in some of their statements, since they weren’t evaluated in this study, not allowing clear conclusions about them. Moreover, in some specific aspects there was missing information regarding the comparison with one of the periods of time, which is important for the conclusion (please see below the specific comments).

Specific comments:

-          Lines 70-82: The inclusion criteria are rewritten below in the form of exclusion criteria. From my point of view, it is not necessary to repeat this information. I would suggest the authors to clarify the inclusion and exclusion criteria. A suggestion would be to consider as inclusion criteria: patients undergoing gastric surgery with an anatomopathological diagnosis of malignancy and the ones with a negative result in the RT-PCR test conducted upon hospitalization. The other criteria mentioned about previous Covid-19 infection and symptoms can be consider as exclusion criteria (lines 78-81).

-          Lines 96-97: clinical and pathological TNM were considered or just pathological TNM – please clarify this point.

-          Lines 129-130: what’s the percentage of patients requiring emergency surgical intervention in the period of 2018-2019? I suggest the authors to consider write all the information concerning surgery, type of surgery, duration of surgery in a table format in order to clarify the results.

-          Lines 149-151: it would also be important to know the mean value of the Charlson Comorbidity Index in the 2018-2019 period.

-          Lines 162-163: It would be important to know the p value when comparing the rate of recurrence between 2016-2017 and 2020-2021.

-          Lines 173-176: Did the authors adjusted these results for possible confounding variables such as duration of surgery? It would be interesting to know the p value when comparing the 2018-2019 period with 2020-2021.

-          Lines 180-183: please clarify the cause of dead of these patients: postoperative infection, other complication during hospitalization or other causes?

-          Lines 245-246: Do you have a literature reference for that information? The motives that lead to fewer patients from rural areas recur to hospital care during pandemic period were not evaluated in this study. They could be possible reasons for this result, but authors can’t draw that conclusion from this study.

-          Lines 253-255: Please review this statement. Between the previously compared periods (2016-2017 vs 2020-2021) similar rates of upper gastrointestinal bleeding (UGB) were observed (27.1% vs 28.6%) and the comparison between the 3 periods regarding UGB was not statistically significat.

-          Lines 264-265: Please consider mention that in this study there were no statistically difference between the 3 periods regarding UGB, decline in body weight and anemia. This is a relevant information that is missing. Authors can also point out possible reasons for these results.

-          Line 267: Please correct this information – the comparison wasn’t done with “previous periods”, but just the period of 2016-2017. Please correct that or add the comparison with 2018-2019 period.

-          268-272: Please add Literature reference for that information. The authors should consider if they want to maintain this statement, since in this work it was not evaluated the relation between Charlson index and severity of symptoms.

-          Lines 273-274: Please review this conclusion, the difference between recurrence rates among the 3 periods was not statistically different. Moreover, it should be also emphasized the small sample size regarding recurrence rates and how it could influence statistic power.

-          Lines 275-277: Authors can’t conclude that, it was not evaluated in this work.

-          Lines 294-296: Care should be taken in drawing these conclusions. In this study, we observe a trend towards a decrease in post-surgical and total hospitalization time over the years and not just during the pandemic period.

-          Lines 320-322: There are no data concerning mean value of the Charlson Comorbidity Index in the 2018-2019 period. Authors should add it or to specify that this conclusion is made based only in the comparison between 2016-2017 period and 2020-2021.

-          Lines 333-336: It should be added as a limitation of the study the small sample size.

-          Lines 350-351: In order to conclude that authors should indicate in the results topic the p value of the comparison between the 2018-2019 period and 2020-2021 regarding total and postoperative hospitalization time.

Author Response

Dear Reviewer, thank you very much for your specific comments and suggestions. We appreciate that you have carefully studied our article, and we hope that we have addressed your points in an accurate manner.

Specific comments

  1. Lines 70-82: The inclusion criteria are rewritten below in the form of exclusion criteria. From my point of view, it is not necessary to repeat this information. I would suggest the authors to clarify the inclusion and exclusion criteria. A suggestion would be to consider as inclusion criteria: patients undergoing gastric surgery with an anatomopathological diagnosis of malignancy and the ones with a negative result in the RT-PCR test conducted upon hospitalization. The other criteria mentioned about previous Covid-19 infection and symptoms can be consider as exclusion criteria (lines 78-81).

Answer : Thank you very much for your suggestion, we addressed this issue as recommended.

  1. Lines 96-97: clinical and pathological TNM were considered or just pathological TNM – please clarify this point.

Answer: We took into consideration your suggestion and we mentioned that in this study we took into consideration both clinical and pathological TNM

  1. Lines 129-130: what’s the percentage of patients requiring emergency surgical intervention in the period of 2018-2019? I suggest the authors to consider write all the information concerning surgery, type of surgery, duration of surgery in a table format in order to clarify the results

Answer: We added the percentage from the 2018-2019 period, and made another table (Table V) in order for the results to be easier to follow.

  1. Lines 149-151: it would also be important to know the mean value of the Charlson Comorbidity Index in the 2018-2019 period

Answer: We made the requested modification

  1. Lines 162-163: It would be important to know the p value when comparing the rate of recurrence between 2016-2017 and 2020-2021.

Answer: We made the requested modification

  1. Lines 173-176: Did the authors adjusted these results for possible confounding variables such as duration of surgery? It would be interesting to know the p value when comparing the 2018-2019 period with 2020-2021.

Answer: Your intuition was right, so, an association was observed between the duration of the surgical intervention and the need for postoperative transfusions (p = 0.029). Patients under-going longer surgical procedures (M = 206.43 min) had a higher likelihood of requiring transfusions compared to those who did not require transfusions (M = 175.88 min).

  1. Lines 245-246: Do you have a literature reference for that information? The motives that lead to fewer patients from rural areas recur to hospital care during pandemic period were not evaluated in this study. They could be possible reasons for this result, but authors can’t draw that conclusion from this study.

Answer: The fact that patients with gastric cancer from rural areas seek medical care in lower proportions is a well-known aspect globally. This can be attributed to various psychosocial factors as well as the level of education, among other factors. However, we have managed to highlight two studies [23-24] that support this idea, especially during the pandemic period.

  1. Lines 253-255: Please review this statement. Between the previously compared periods (2016-2017 vs 2020-2021) similar rates of upper gastrointestinal bleeding (UGB) were observed (27.1% vs 28.6%) and the comparison between the 3 periods regarding UGB was not statistically significant.

Answer: Indeed, there are not differences between the first period and the pandemic period. However, we highlighted the differences between the second period and the pandemic period. ,,This trend can be ultimately justified by a higher proportion of patients presenting with upper gastrointestinal hemorrhage 28.6% (pandemic period) compared to 18% (2018-2019) even though in the first period of the study the percentage (27.1%) was similar to that of the pandemic period. “

  1. This trend can be ultimately justified by a higher proportion of patients presenting with upper gastrointestinal hemorrhage 28.6% (pandemic period) compared to 18% (2018-2019) even though in the first period of the study the percentage (27.1%) was similar to that of the pandemic period

Answer: We made the requested addition

  1. Line 267: Please correct this information – the comparison wasn’t done with “previous periods”, but just the period of 2016-2017. Please correct that or add the comparison with 2018-2019 period.

Answer: We made it clear that there were significant differences between the pandemic period and the 2016-2017 period. ,,Last but not least, patients undergoing surgery during the pandemic exhibited a statistically significant higher mean Charlson index (p=0.024) compared to 2016-2017, with an average of 6.0±2.88 compared to 4.74±2.04. Although there was no statistically significant difference between the pandemic period and the 2018-2019 period when comparing the mean Charlson index, it is worth noting that an increase from 5.44±2.49 to 6.0±2.88  was observed in the most recent period.”

  1. 268-272: Please add Literature reference for that information. The authors should consider if they want to maintain this statement, since in this work it was not evaluated the relation between Charlson index and severity of symptoms.

Answer: We removed this statement

  1. Lines 273-274: Please review this conclusion, the difference between recurrence rates among the 3 periods was not statistically different. Moreover, it should be also emphasized the small sample size regarding recurrence rates and how it could influence statistic power.

Answer: We modified it so it is clearly mentioned between which periods are the significantly differences present. ,, In addition to this, a statistically significant higher proportion of patients experi-encing recurrence was observed (p<0.05) between the 2018-2019 and the pandemic pe-riod, with an increase from 4% to 17.9%. This result should be interpreted with caution, taking into consideration the sample size of the patient cohort and the fact that when comparing all three periods, the result does not generate statistically significant dif-ferences between them.”

  1. Lines 275-277: Authors can’t conclude that, it was not evaluated in this work

Answer: We agree, as a consequence, we removed that statement.

  1. Lines 294-296: Care should be taken in drawing these conclusions. In this study, we observe a trend towards a decrease in post-surgical and total hospitalization time over the years and not just during the pandemic period.

Answer: Indeed a trend towards a decrease is observed. However, a statistically significant difference is observed for 2 variables only when the pandemic period is taken into consideration. We believe that the pandemic period has accelerated even more this trend of decreasing the hospital duration, this is supported by studies worldwide [9,7,28].

  1. Lines 320-322: There are no data concerning mean value of the Charlson Comorbidity Index in the 2018-2019 period. Authors should add it or to specify that this conclusion is made based only in the comparison between 2016-2017 period and 2020-2021.

Answer: We added data for 2018-2019 period. Furthermore, the Cox regression analysis, which examined the risk of postoperative mortality, various parameters were considered throughout the 6-year study period. This regression analysis revealed that the Charlson index is a risk factor for mortality in patients with gastric cancer over the entire 6-year period. Considering that the average value of the Charlson index was highest during the pandemic period, significantly higher compared to the 2016-2017 period, we consider it to be an important factor for patient prognosis. The purpose of this Cox analysis is not to highlight the variation of the Charlson index over time but rather to demonstrate its actual effect on mortality risk.

  1. Lines 333-336: It should be added as a limitation of the study the small sample size.

Answer: We made it clear that the sample size is an important limitation of this study. ,, Firstly, it was conducted in a single clinic of a University Tertiary hospital in Romania, and therefore the size of the patient cohort should be taken into consideration , which limits the generalizability of the findings to a national level”

  1. Lines 350-351: In order to conclude that authors should indicate in the results topic the p value of the comparison between the 2018-2019 period and 2020-2021 regarding total and postoperative hospitalization time.

Answer: We made the modification in the results part, and presented the p value for the requested periods. However, since the differences were statistically significant only between 2016-2017 and the pandemic period we preferred to remove the statement from the conclusions.

Reviewer 2 Report

The authors present an interesting paper comparing patterns of surgical use for gastric cancer in the years leading up to the COVID-19 pandemic versus the years at the beginning of the pandemic. The authors provide a background section which gives necessary rationale for their work, and their methods section is sufficiently detailed for the study to be replicated. My comments and suggestions for revision are included below.

1. The authors restricted their sample to only those patients with no previous SARS-CoV-2 infection. However, the authors examine surgeries completed from 2/26/2020-10/1/2021. By 10/2021, there had already been several waves of infection and it would be expected that a large proportion of the population may have had a SARS-CoV-2 infection by this point. Moreover, those who were unable to work remotely, which was largely lower income, lower SES people, would have been more likely to have had a previous infection and not be eligible for this research. This choice may introduce substantial selection bias into the authors research, as it is likely that lower income, lower SES people may have different surgical outcomes than higher income people. In addition, the authors compare crude rates of surgery before and after the first case of SARS-CoV-2, but do not account for the fact that their crude pandemic rate may be substantially lower than the actual rate, as a large number of surgical recipients may be omitted from this rate due to having a previous SARS-CoV-2 infection. These implications must be addressed in this research.

2. Throughout the results section p-values presented in written results do not match p-values presented in tables. This needs to be edited for consistency.

3. The written results are very confusing and hard to follow because there are no tables which present the authors’ main findings. Tables 1-4 can logically be combined into a single Table 1 describing characteristics of the study cohort by surgical year. Subsequently, additional tables should be added which summarize all results discussed in the written results.

4. Much of the findings presented by the authors are purely descriptive, despite availability of variables to conduct multivariable analysis controlling for patient characteristics such as age, gender, environment, and cancer stage, etc. This research would greatly benefit from a revision to the methods to conduct multivariable analyses of associations with the authors main outcomes (ie is surgical year associated with each surgical outcome, controlling for individual patient characteristics?).

There are a few typos throughout the manuscript. Please perform a spell check before finalizing. 

Author Response

Dear Reviewer, thank you very much for taking your time and analyzing our work. We appreciate that you have carefully studied our article, and we hope that we have addressed your points in an accurate manner.

Specific comments

  1. The authors restricted their sample to only those patients with no previous SARS-CoV-2 infection. However, the authors examine surgeries completed from 2/26/2020-10/1/2021. By 10/2021, there had already been several waves of infection and it would be expected that a large proportion of the population may have had a SARS-CoV-2 infection by this point. Moreover, those who were unable to work remotely, which was largely lower income, lower SES people, would have been more likely to have had a previous infection and not be eligible for this research. This choice may introduce substantial selection bias into the authors research, as it is likely that lower income, lower SES people may have different surgical outcomes than higher income people. In addition, the authors compare crude rates of surgery before and after the first case of SARS-CoV-2, but do not account for the fact that their crude pandemic rate may be substantially lower than the actual rate, as a large number of surgical recipients may be omitted from this rate due to having a previous SARS-CoV-2 infection. These implications must be addressed in this research.

Answer: Thank you very much for your comment. Indeed, you have highlighted certain aspects quite well. The pandemic situation has evolved in waves as you have described, so it is possible that there could have been COVID-19 positive patients who were not included in our study. The question is how many were in this situation and whether their number would have significantly influenced the study results. To answer this question, we need to consider the situation from a broader perspective and look at the bigger picture.

Firstly, this decrease in surgical interventions has been reported worldwide, across all continents, especially elective surgeries that were postponed in the initial phase of the pandemic. Furthermore, if we take a closer look at the results of our study, you will notice that during the pandemic period, 90% of the patients were in stage III or IV. The limited number of patients in stage II (no patients in stage I) is primarily due to reduced screening for gastric cancer during this period, with some countries reporting up to an 80% decline [8, 17-22].

Moreover, the symptoms of patients in the early stages are less intense, and therefore they did not seek medical services for various reasons mentioned in both the study and the specialized literature. Along with the advanced stage at presentation, their symptomatology, particularly in the case of emesis, was significantly higher. What we want to emphasize after considering these aspects and psychosocial factors is the fact that the smaller sample size during the pandemic is mainly due to these factors, as mentioned in the literature, and not due to the number of COVID-19 positive cases with gastric cancer that were not included.

To assume that the proportions of patients would be approximately similar to previous periods, we would have to speculate that a large portion of patients with stage I or II gastric cancer had COVID-19 and therefore did not got included in our study. This is false, especially considering that patients in more advanced stages of the disease have a less potent immune system and are more susceptible to developing SARS-CoV-2 infection. Therefore, if SARS-CoV-2 infection were the main factor leading to the decrease in the sample size, we would have observed a decrease in the number of patients in stages III and IV (being COVID-19 positive) and not the other way around. 

  1. Throughout the results section p-values presented in written results do not match p-values presented in tables. This needs to be edited for consistency.

 Answer: We tried to identify  all these situations mentioned and managed to address them. We just want to point out that we did conduct statistical analysis for variables between the 3 periods of time and individual comparison between the pandemic period and the first period, respectively the pandemic period and the 2018-2019. Therefore, there are some different p values regarding the analyzed data

  1. The written results are very confusing and hard to follow because there are no tables which present the authors’ main findings. Tables 1-4 can logically be combined into a single Table 1 describing characteristics of the study cohort by surgical year. Subsequently, additional tables should be added which summarize all results discussed in the written results.

Answer: We did took into consideration and even made a table with all variables but we had to give up on that option since it would have been a 3 and half page long table. Moreover there are some correlations and associations between variables and it would have been even harder to put them all together at the end of that comprehensive table. However, we appreciate your comment, and indeed we added information from the text in the already existing tables and made another table for variables such as duration of hospitalization surgery duration, etc. All the extra table results are the ones that present statistical results which cannot be included in a typical table from our point of view.

  1. Much of the findings presented by the authors are purely descriptive, despite availability of variables to conduct multivariable analysis controlling for patient characteristics such as age, gender, environment, and cancer stage, etc. This research would greatly benefit from a revision to the methods to conduct multivariable analyses of associations with the authors main outcomes (ie is surgical year associated with each surgical outcome, controlling for individual patient characteristics?).

Answer: As we all now, there are few studies that present the full consequences of the Covid-19 pandemic on gastric cancer patients. It is true that most of our work is descriptive because we want to draw attention to the way things have evolved during this pandemic. Advanced statistical analysis reveals significant findings such as the correlation between surgical duration and hospitalization length, the association between the need for transfusion and surgical duration, or the presence of emesis at admission associated with more advanced disease stage and TNM, among others. Moreover, the Cox regression presented that the risk for postoperative death was increased by Charlson index and type of surgery.  All these parameters have undergone considerable changes during the pandemic compared to previous periods, thus these prognostic factors have had a more significant impact on patient prognosis during this unprecedented period in recent history. As for the future, our aim is to monitor the way things will unfold in the upcoming periods, as it is expected that patients who did not seek hospital care due to the reasons mentioned in the study may present in larger numbers and potentially at more advanced stages. This is merely a hypothesis at this point, but we are keenly interested in observing how things will progress.

Round 2

Reviewer 1 Report

Dear authors, I congratulate you on the remarkable work on this manuscript.

From my point of view, the topics Methods and Results are clearly described and presented. In addition, the discussion is well structured and highlights the most important topics.

Regarding lines 246-247, I agree with you that is well known that patients with gastric cancer from rural areas seek medical care in lower proportions. However, the reasons for that happening in your hospital area during the pandemic period were not evaluated in this study. Therefore, you can point out some probable reasons for this, but not name precisely what those reasons were,  

Author Response

Thank you very much for the kind words.

We have made adjustments, mentioning that the reasons listed are probable rather than certainties.

Best regards,

Ms. Faur

Reviewer 2 Report

Thank you to the authors for responding to my concerns, my remaining concerns are below.

1. The authors present a well-informed basis for why their study may not have missed recruitment of a large number of patients due to previous SARS-CoV-2 infection. However, the concern of selection bias due to this fact has not been addressed and remains a large source of potential bias in this observational study. This potential bias should be recognized as a limitation of the study. Selection bias does not concern whether or not the authors “missed” representative cases of cancer (this is more a concern of external validity of the research), but whether or not factors potentially associated with the outcome of the research (cancer treatment) were also associated with probability of participation in the research.

2. It is still unclear why p-values reported in the text do not match those in the tables. In particular, the p-value reported on line 122 does not match the p-value listed in Table 1. The p-value listed on line 131 does not match that in Table 2. The p-value listed on line 156 does not match that presented in Table 3. The p-value listed on line 163 does not match that presented in Table 3, etc.  

3. It remains very confusing to read through the results without results organized in any manner or in any tables. Perhaps subheadings may help to organize this section if the authors do not wish to present their results in a table. After reflecting on why p-values do not match between text and tables, it becomes very confusing as to why there is seemingly no overlap between written results and results presented in tables.

4. The authors suggest that a descriptive analysis is the most methodologically sound way to demonstrate how “things have evolved during the pandemic”. A multivariable analysis analyzing associations between time and surgical outcomes, controlling for potential confounding factors, would be a more appropriate way to definitively summarize these changes as it would account for potential confounding bias. The authors should seriously consider the inclusion of such an analysis, as descriptive findings are otherwise unreliable due to the influence of potential confounding.

Author Response

Thank you very much for your comments. We hope that this time we have been clearer and managed to identify and address your requirements.

  1. The authors present a well-informed basis for why their study may not have missed recruitment of a large number of patients due to previous SARS-CoV-2 infection. However, the concern of selection bias due to this fact has not been addressed and remains a large source of potential bias in this observational study. This potential bias should be recognized as a limitation of the study. Selection bias does not concern whether or not the authors “missed” representative cases of cancer (this is more a concern of external validity of the research), but whether or not factors potentially associated with the outcome of the research (cancer treatment) were also associated with probability of participation in the research.

Answer: We included this limitation as well ,, Another limitation of the study is the potential for selection bias due to previous SARS-CoV-2 infection and multiple waves of infection. This could lead to a disproportionate representation of lower-income individuals unable to work remotely, affecting generalizability and representativeness. “

  1. It is still unclear why p-values reported in the text do not match those in the tables. In particular, the p-value reported on line 122 does not match the p-value listed in Table 1. The p-value listed on line 131 does not match that in Table 2. The p-value listed on line 156 does not match that presented in Table 3. The p-value listed on line 163 does not match that presented in Table 3, etc. 

Answer: We apologize, but there seems to be a slight misunderstanding regarding this aspect. The p-values you mentioned do not appear in the tables at all. They are mentioned only in the text because they represent the results of analyzing variables other than those presented in the tables. For example, in row 122, "The statistical analysis and application of the chi-square test to detect differences in proportions among the three periods resulted in a p-value of <0.001." This p-value does not appear in Table 1 because Table 1 does not present an analysis of the total number of interventions performed in each period. Furthermore, in row 131, "After statistical analysis and the application of the chi-square test, a p-value of 0.255 was obtained, indicating no significant difference among the three study periods." Again, here we are discussing the number of emergency surgical interventions performed over the three periods, which is not presented in Table 2, but rather the clinical manifestations of patients, and so on. In row 163, "Furthermore, a significant increase in recurrence was observed in the pandemic period of the study compared to the 2018-2019 period (p<0.05)." Here, we are referring to a p-value that involves an individual analysis of the pandemic period compared to the 2018-2019 period, while the table presents a p-value indicating the result of the analysis of all three periods together... There may be a misunderstanding, but our team has made sure multiple times that the data aligns. Please take another look to ensure that we are all discussing the same thing.

  1. It remains very confusing to read through the results without results organized in any manner or in any tables. Perhaps subheadings may help to organize this section if the authors do not wish to present their results in a table. After reflecting on why p-values do not match between text and tables, it becomes very confusing as to why there is seemingly no overlap between written results and results presented in tables.

We agree, and this is due to the extensive results we obtained. We decided to divide the results section to make it easier to follow. However, the magnitude of the results limits our presentation method. Thus, under each table, we have included the individual differences between the pandemic period and the first or second period of the study, as well as the associations between variables. If we had listed these two "concepts" separately, we believe (though we may be mistaken) that it would have been much more challenging. We hope that the modifications made meet your requirements.

  1. The authors suggest that a descriptive analysis is the most methodologically sound way to demonstrate how “things have evolved during the pandemic”. A multivariable analysis analyzing associations between time and surgical outcomes, controlling for potential confounding factors, would be a more appropriate way to definitively summarize these changes as it would account for potential confounding bias. The authors should seriously consider the inclusion of such an analysis, as descriptive findings are otherwise unreliable due to the influence of potential confounding.

Upon further analysis of your comments, we would like to apologize for the misunderstanding. We do not consider a descriptive analysis to be the most methodologically sound way to demonstrate how "things have evolved during the pandemic." We understand your point that our study incorporates numerous variables, making it challenging to track them effectively. Certainly, the obtained results are not absolute, and a multivariable analysis analyzing associations would indeed improve the quality and outcomes of this article. However, implementing such an analysis would require a fundamental rework and significant restructuring of the entire article. In addition to the limitations you have already mentioned in your previous comments, we have made an effort to present the situation and compensate through other aspects, including the evaluation of multiple parameters, to the best of our abilities. We appreciate your understanding regarding the limitation in our study, and we hope you can consider other aspects through which we have attempted to compensate for this shortfall.  

We hope that we have addressed all of your comments and concerns to the best of our ability.